# Comparison of Safety and Efficiency between Tiger-2 Catheter with Right Radial Artery Access and Judkins Catheter with Left Radial Artery Access

**DOI:** 10.3390/jcm10174020

**Published:** 2021-09-06

**Authors:** Katarzyna Klimek, Mateusz Świątek, Konrad Klocek, Michał Tworek, Maciej Zwolski, Krzysztof Milewski, Adam Janas

**Affiliations:** 1Center of Cardiovascular Research and Development American Heart of Poland, 40-028 Katowice, Poland; klimekkatarzyna59@gmail.com (K.K.); konrad.klocek23@gmail.com (K.K.); michal.tworek.med@gmail.com (M.T.); maciejzwolski97@gmail.com (M.Z.); 2Faculty of Medical Sciences in Katowice, Medical University of Silesia, 40-055 Katowice, Poland; 3The Jerzy Kukuczka Academy of Physical Education, 40-959 Katowice, Poland; k.milewski@ahp-ccrd.org; 4Andrzej Frycz Modrzewski Kraków University, 30-075 Kraków, Poland; adamjjanas@gmail.com

**Keywords:** catheter, coronary angiography, contrast volume, radiation exposure, radial access

## Abstract

We sought to compare the safety and efficiency of Tiger-2 in the right radial and Judkins catheter in the left radial access. We retrospectively collected data of 487 patients, involving 172 patients after coronary angiography with Judkins on the left radial artery and 315 patients with Tiger-2 on the right radial artery access. There were no differences in baseline characteristics, except for hypertension ratio and mean age. There was a difference in pulse absence on the radial artery. The volume of contrast used was higher in the Judkins group. Both groups differed in the amount of drugs administered (NTG and heparin). Fluorescence times were comparable between groups. Radiation dosage and AK was significantly greater in the Tiger-2 group. The Tiger-2 catheters were significantly more often changed to another type of catheter (100 changes) than the Judkins (12 changes). However, there was no statistical difference in access site change. Judkins with left radial access seems to be a safer option because of the lower radiation exposure and less incidence of complications than Tiger-2 with right radial access, however, it requires a higher volume of contrast.

## 1. Introduction

In 1960 F. Mason Sones performed the first selective Coronary Angiography (CAG), however, from that moment on the technique of coronary artery imaging has dynamically improved. Nowadays CAG is a gold standard tool for myocardial infarction assessment and for coronary artery diseases treatment [1]. Radial artery access is preferred rather than the transfemoral approach due to lower bleeding complications ratio and reduction in costs and hospitalization time [1]. In addition, there are no major nerves or vessels located close to the artery, which decreases the risk of complications. 

Despite these benefits, the transradial approach is more difficult to master than transfemoral access [2]. However, the choice between right or left radial artery depends on the decision determined by an operator [3]. According to studies, the majority of them prefer the right approach because of their proficiency in catheter manipulation and simplicity with access setup [4]. Nonetheless, the left radial access provides a straighter road to the coronary arteries than the right. It is caused by avoiding the brachiocephalic trunk which can be very twisty especially in the elderly and in patients with poorly controlled hypertension [5]. Therefore, comparisons between right and left radial artery access are scarce and may lead to different results. Although a meta-analysis performed by Shah et al. [5], suggests a difference in terms of the volume of the used contrast and the fluoroscopy time which favors the left radial approach. Not only the access site but also the type of diagnostic catheter may play role in the safe and efficient performance of CAG. Another possible option is the distal transradial access in the anatomical snuffbox. Its benefits are lower risk of ischemia, and better patient comfort [6].

A comparison between the two most commonly used types of catheter, described in the JUDGE study, proved the superiority of the Tiger-2 catheter over the Judkins catheters in terms of the volume of used contrast, fluoroscopy time and spasm occurrence- nevertheless, the right radial artery (RRA) was chosen as the obligatory access [5]. Langer et al.’s [7] comparison study also confirmed these data in a field of contrast and fluoroscopy time. The percentage of forearm hematomas was also decreased in the Tiger group.

Nevertheless, there were no studies that put together both different access sites and types of catheter. The purpose of this study was to compare safety and efficiency between the Tiger-2 catheter via RRA and the Judkins catheter on left radial artery (LRA) access in patients undergoing CAG (Figure 1). 

## 2. Methods

This was a retrospective study based on data collection of 487 patients between 2018 and 2019 who underwent CAG in five cardiac centers located in Tychy, Bielsko-Biała, Dąbrowa Górnicza, Chrzanów and Ustroń. All procedures in every cardiac hospital were performed by an experienced, board-certified interventionalist. Consecutive patients undergoing non-urgent CAG were enrolled in this registry. All patients received 50 IU/kg of heparin or a 5000 IU bolus. Intra-arterially nitroglycerine was administrated on the operator’s directions. 

Patients were excluded from the registry if they presented with an indication for urgent CAG (ST-elevation myocardial infarction or non-ST-elevation myocardial infarction with an indication for CAG within two hours), hemodynamic instability, non-palpable radial artery, prior coronary artery bypass grafting, dialysis, or known severe renal impairment (creatinine clearance < 30 mL/min). Patients were also excluded in the case of non-availability of either the study catheter or qualified operator (fully licensed interventional cardiologist).

The choice between right and left Judkins depends on the access site (right of left). The operator can choose between Judkins and Tiger catheters according to the conditions of the intervention. 

The registry only included an angiogram with at least four standard views to image the left coronary artery (LCA) and two standard views for the right coronary artery (RCA).

### 2.1. Endpoints

The objectives of the study were to compare the performance of Tiger-2 via RRA vs. Judkins via LRA in terms of catheter failure, contrast volume, duration of CAG, fluoroscopy time and dose area product (DAP: mGy*cm^2^) and pulse absence during discharge.

### 2.2. Definitions

Catheter failure was defined as the need for study catheter change and/or access site change for completion of CAG. Clinically significant hematomas were defined as any hematoma which led to pain lasting at least 12 h or hand functional deficiency. Significant spasm was defined as leading to catheter failure. Angiography time was defined as the interval from the insertion of the first diagnostic coronary catheter in the sheath until the exit of the last diagnostic catheter from the sheath. Fluoroscopy time was defined as the cumulative duration of fluoroscopy during angiography time.

### 2.3. Statistical Analysis

Categorical data are presented as frequencies and group percentages and compared with Fisher’s exact test, whereas continuous data with skewed distribution were presented as means with standard deviations (SD) and compared with the Mann-Whitney U test. Statistical significance was considered for *p*-values < 0.05. The analysis was performed with GraphPad Prism 8.

## 3. Results

The study evaluated data from a total of 487 patients. CAG by the Tiger-2 catheter with right radial artery access was performed on 315 (64.6%) patients, whereas 172 (35.4%) patients underwent CAG with Judkins catheters via left radial artery access. A comparison of baseline characteristics is presented in Table 1. 

There were no differences between the two groups except for mean age (*p* = 0.02). The patients in the Tiger-2 and Judkins groups were well-matched for clinical conditions: diabetes mellitus, dyslipidemia, smoking and atrial fibrillation. However, the presence of hypertension was more frequent in Judkins left group: 87.8% (151) vs. Tiger-2 right group 74.9% (236) (*p* > 0.0007). 

Unsuccessful cannulation with Tiger-2 occurred more frequently when compared to the Judkins approach 100 (31.75%) vs. 12 (6.98%), *p* < 0.0001). Catheter failure led to the replacement of Tiger-2 by left, right, or both Judkins catheters (Figure 2). However, the difference in access site change from radial to femoral was not significant.

The vast majority of complications occur after using the Tiger-2 catheter and the difference between the Tiger-2 and Judkins groups presents mainly as a lack of pulse in the radial artery while patient’s discharge (*p* = 0.0006) (Figure 3).

Patients who underwent CAG with Judkins by left radial access received a higher amount of nitroglycerin (*p* = 0.0064) than patients from the Tiger-2 group (Figure 4). Similarly, a greater volume of contrast was used to demonstrate the coronary arteries with Judkins and left radial approach than Tiger-2 and the right radial (*p* < 0.0001).

There were no statistically significant differences in fluoroscopy time in both the Judkins and the Tiger-2 group, although patients with Tiger-2 right radial access were exposed to a much larger number of DAP (*p* = 0.0096) and Air KERMA (*p* = 0.0196).

## 4. Discussion

The transradial approach is recommended as the gold standard approach for coronary interventions. The catheters used for the femoral approach are recommended. Special multipurpose catheters for transradial access like Tiger-2 are suggested as further options [8]. So far no study has compared the performance of Judkins catheters from the left radial artery and Tiger-2 from the right radial access. This is the first study in which those comparisons were performed.

Our results showed that the lack of pulse during discharge occurs more often in the group where Tiger-2 was used. This could be caused by the lower dosage of heparin utilization in this group. Moreover, engaged ostium with Tiger-2 requires more catheter manipulation. It could lead to endothelial irritation vasospasm and subsequently arterial injury which leads to radial closure. Our observations are opposite to other studies where Tiger-2 caused less spasm and injury than Judkins utilized from the right radial artery [5]. In a previous study, it had been shown that a shorter procedure time leads to less radial occlusion [9,10]. However, in this study, the time of the procedure was comparable between groups. Also, significantly more frequent changes of Tiger-2 for other catheters could lead to radial occlusions. Besides that, there were no differences in safety endpoints.

When comparing the volume of contrast medium usage, Tiger-2 was significantly better than Judkins from the left radial. This was consistent with results from other studies where Judkins used from the right radial was compared to Tiger-2 [11]. It might be caused by Tiger-2’s better performance in reaching the coronary arteries. Moreover, the fluoroscopy time was similar between the groups and to other studies [12]. Nevertheless, the radiation dose was lower in the Judkins catheter group than in Tiger-2. The explanation could be that Tiger-2 was significantly frequently changed to other catheters which lead to an increase in radiation dose.

The need for catheter changes was in favor of the Judkins group. Tiger-2 was changed in more than 30% of cases. This could have led to more frequent radial occlusions and radiation dose increases. The need for Tiger-2 frequent changes was rather comparable with similar studies [13]. However, different approaches may have also been beneficial, like the distal transradial approach that had to be changed in only 10% of cases [2].

### Limitations

The main drawbacks of this work are those inherent to any observational study [14]. The total amount of contrast used (including leftovers) was studied; therefore, no assumption could be made regarding the contrast volume received by the patient. The TIG catheter was the only analyzed multifunctional catheter in this study. The result of our study does not apply to other multifunctional radial catheters (e.g., Kimny catheter) or to other guiding catheters. The catheters might differ regarding specific design and material. Moreover, the analysis of the consecutive patients does represent the “real world” in large interventional centers. 

## 5. Conclusions

CAG with a multipurpose Tiger-2 catheter was connected with higher radiation dosage, the frequent need of change to other catheters because of unsuccessful cannulation and a significantly higher ratio of radial occlusion. However, CAG performed with Judkins on the left radial approach required more contrast and nitroglycerine usage. Nevertheless, this study is hypothesis-generating only. 

## Figures and Tables

**Figure 1 jcm-10-04020-f001:**
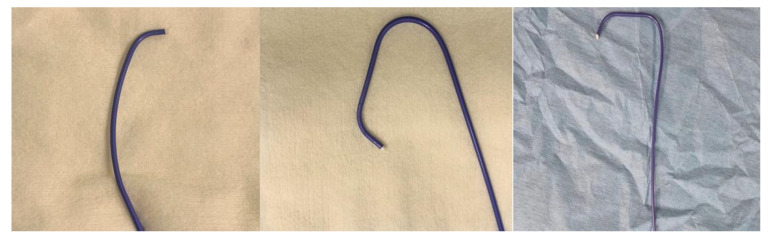
Right Judkins Catheter (**left**), Left Judkins Catheter (**middle**) and Tiger-2 Catheter (**right**).

**Figure 2 jcm-10-04020-f002:**
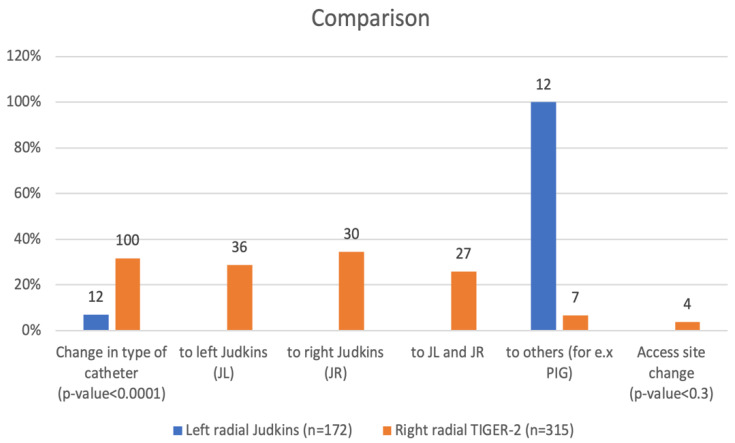
Comparison of change in type of catheters and site changes.

**Figure 3 jcm-10-04020-f003:**
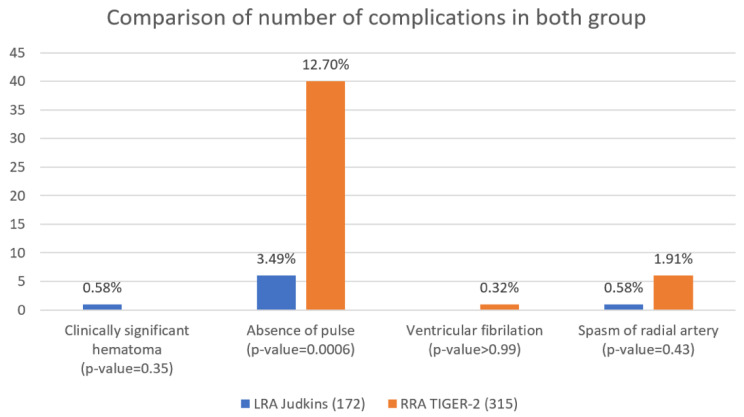
Comparison of complications in both groups.

**Figure 4 jcm-10-04020-f004:**
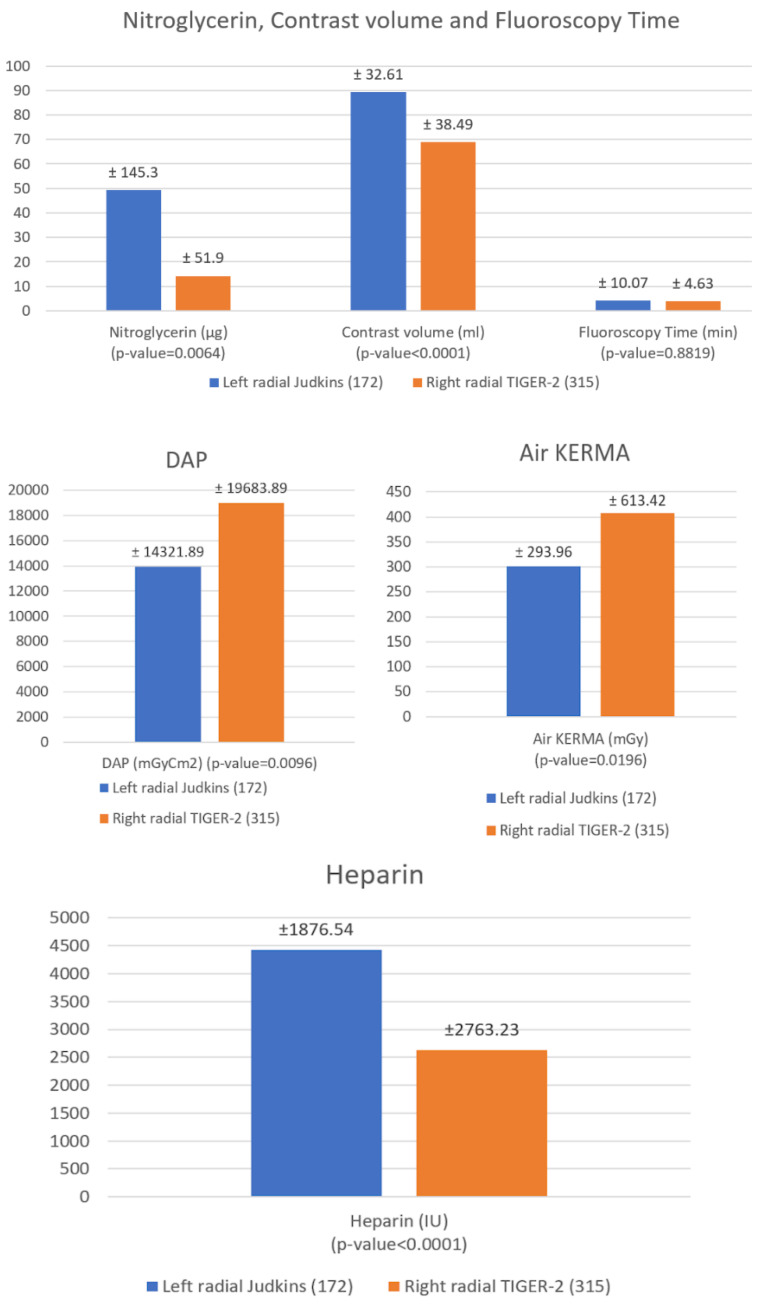
Differences between approaches in Nitroglycerin, Contrast volume, Fluoroscopy time, DAP dosage and Air KERM.

**Table 1 jcm-10-04020-t001:** Baseline characteristics of patients (BMI = Body Mass Index).

	LAR Judkins (172)	RAA TIGER-2 (315)	*p*-Value
**Mean age**	67.66(±10.69)	65.37 (±11.16)	0.02
**Male**	61.05% (105)	60.63% (191)	>0.99
**BMI**	28.39 (±5.11)	28.71 (±4.90)	0.11
**Diabetes mellitus**	30.81% (53)	29.21% (92)	0.76
**Hypertension**	87.79% (151)	74.92% (236)	0.0007
**Dyslipidemia**	47.67% (82)	47.62% (150)	>0.99
**Smoking**	20.35% (35)	15.56% (49)	0.21
**Atrial fibrillation**	13.37% (23)	14.92% (47)	0.69

## Data Availability

The data presented in this study were obtained from internal American Heart of Poland databases, which are restricted.

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
