# Peer review of "Comparison of Safety and Efficiency between Tiger-2 Catheter with Right Radial Artery Access and Judkins Catheter with Left Radial Artery Access"

_jcm, 2021, doi:10.3390/jcm10174020_

Round 1
Reviewer 1 Report
There is growing interest in the left radial approach
Some remarks
- Sometimes the article expires in some technicalities, try to be simpler and clearer
- It is not intuitive for everyone, specify the use of two judkins catheter (right and left)
- insert an example image with judkins and a Tiger-2
- specify the radial experience of the hospital
- If it's not too complicated, report the MATRIX SCORE (EuroIntervention, August 2021)
- I would also mention the possibility of integrating your approach with the distal radial (I would ask you to cite a very innovative article in this regard: PMID 29543187)
Author Response
Response to Reviewer 1 Comments
Point 1: Sometimes the article expires in some technicalities, try to be simpler and clearer
Response 1: Changes were applied in “discussion”, “results” and “methods” paragraphs.
Point 2: It is not intuitive for everyone, specify the use of two Judkins catheter (right and left)
Response 2: Specifications was added to the “methods” paragraph.
Point 3: Insert an example image with Judkins and a Tiger-2
Response 3: Example Images was added to the manuscript.
Point 4: Specify the radial experience of the hospital
Response 4: It was added to the “methods” paragraph.
Point 5: If it's not too complicated, report the MATRIX SCORE (EuroIntervention, August 2021)
Response 5: Unfortunately, reporting the Matrix Score is too complicated to include.
Point 6: I would also mention the possibility of integrating your approach with the distal radial (I would ask you to cite a very innovative article in this regard: PMID 29543187)
Response 6: The distal radial approach was added in “Introduction” and “discussion” paragraphs.
Conclusions and Introduction were improved as advised and English was improved by a native speaker.

Reviewer 2 Report
Generally article is well prepared. The issue of higher amount of contrast used in Judkins group should be commented by authors. What is their explanation of this fact ?
Author Response
Response to Reviewer 2 Comments
Point 1: Generally, article is well prepared. The issue of higher amount of contrast used in Judkins group should be commented by authors. What is their explanation of this fact?
Response 1: The higher amount of contrast used in Judkins group comment were added in “Discussion” section.
